

**Field Investigations of Coastal Sea Surface Temperature Drop**
**after Typhoon Passages**

3              Dong-Jiing Doong [1]*     Jen-Ping Peng [2]    Alexander V. Babanin [3]

[1] Department of Hydraulic and Ocean Engineering, National Cheng Kung University, Tainan,

5        Taiwan

[2] Leibniz Institute for Baltic Sea Research Warnemuende (IOW), Rostock, Germany
[3] Department of Infrastructure Engineering, Melbourne School of Engineering, University of

8        Melbourne, Australia

9    ----

**\*Corresponding author:**
Dong-Jiing Doong
Email: doong@mail.ncku.edu.tw
Tel: +886 6 2757575 ext 63253
Add: 1, University Rd., Tainan 70101, Taiwan

15         Department of Hydraulic and Ocean Engineering, National Cheng Kung University



1          **Abstract**

Sea surface temperature (SST) variability affects marine ecosystems, fisheries, ocean primary
productivity, and human activities and is the primary influence on typhoon intensity. SST drops
of a few degrees in the open ocean after typhoon passages have been widely documented;
however, few studies have focused on coastal SST variability. The purpose of this study is to
determine typhoon-induced SST drops in the near-coastal area (within 1 km of the coast) and
understand the possible mechanism. The results of this study were based on extensive field data
analysis. Significant SST drop phenomena were observed at the Longdong buoy in northeastern
Taiwan during 43 typhoons over the past 20 years (1998~2017). The mean SST drop (ΔSST)
after a typhoon passage was 6.1 °C, and the maximum drop was 12.5 °C (Typhoon Fungwong
in 2008). The magnitude of SST drop was larger than most of the observations in the open ocean.
The mean duration of SST drop was 24 hours, and on average, 26.1 hours were required for the
SST to recover to the original temperature. The coastal SST drops at Longdong were correlated
with the moving tracks of typhoons. When a typhoon passes south of Longdong, the strong and
persistent longshore winds induce coastal upwelling and pump cold water up to the surface,
which is the dominant cause of SST drops along the coast. In this study, it was determined that
cold water mainly intruded from the Kuroshio subsurface in the Okinawa Trough, which is
approximately 50 km from the observation site. The magnitude of coastal SST drops depends
on the area of overlap between typhoons generating strong winds and the Kuroshio. The dataset
used in this study can be accessed by https://doi.pangaea.de/10.1594/PANGAEA.895002.
Keywords: Coastal SST drop, Typhoon, Upwelling, Kuroshio, Data buoy
**1. Introduction**
Similar to the Earth's atmosphere, sea surface temperature (SST) changes diurnally, but the
range is small. Significant SST drops (ΔSST) after typhoon (hurricane) passages have been

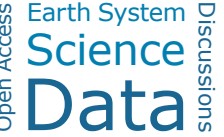

widely known and reported in the world's oceans, including the Northwest Pacific (Sakaida et
al., 1998; Tsai et al., 2008a, 2008b, 2013; Chen et al., 2003; Wada et al., 2005, 2009; Chang et
al., 2008; Wu et al., 2008; Morimoto et al., 2009; Hung et al., 2010; Kuo et al., 2011; Sun et al.,
2015; Subrahmanyam, 2015), Northeast Pacific (Bingham, 2007), India Ocean (Rao et al., 2004;
Gopalakrishna et al., 1993), and South China Sea (Shang et al., 2008; Jiang et al., 2009; Tseng
et al., 2010; Chiang et al., 2011). SST drops are larger in scale following a typhoon passage
than under regular temperature variability and may affect marine ecosystems and the primary
productivity of the ocean (Lin et al., 2003b; Siswanto et al., 2007). Cold water increases
nutrients for marine life. Several studies (Babin et al., 2004; Hanshaw et al., 2008; Liu et al.,
2009; Kawai and Wada, 2011, Cheung et al., 2013; Xu et al., 2017) have reported that
chlorophyll-*a* increases when SST drops after the passages of tropical cyclones. In contrast, fish
species that cannot tolerate cold may die if the water temperature drops dramatically over a
short period of time. In addition, the water temperature has a major impact on human comfort
and safety during swimming, surfing, and snorkeling activities.
Upwelling and entrainment (vertical mixing) have been identified as the main causes of sea
surface water temperature cooling after a typhoon passage (Price, 1981; Rao et al., 2004;
Narayan et al., 2010; Shen et al., 2011; Chen et al., 2012). The maximum SST drop caused by
typhoons rarely exceeds 6 °C (Wentz et al., 2000). Price (1981) presented SST drops of 3 °C
and 1 °C in US waters during Hurricane Eloise in 1975 and Hurricane Belle in 1976,
respectively. He noted that the SST decrease beneath a moving hurricane was mainly caused by
entrainment and that the heat changes in the air and sea play minor roles. Stronger wind stress
and the associated curl surface wind trigger more substantial ocean mixing and induce the
mixing of sea surface water with colder and deeper waters. Wada et al. (2009) studied the role
of vertical turbulent mixing (VTM) in sea surface cooling during typhoon Rex in 1998 in the
Northwestern Pacific Ocean near Japan, during which the SST dropped by nearly 3 °C. They



concluded that sea surface cooling was caused by shear-induced VTM during the fast-moving
phase of the typhoon; in contrast, sea surface cooling was caused by Ekman pumping during
the slow phase of the typhoon. Notably, unless the waters are very shallow, the wind-mixing
mechanism usually occurs through the action of wind-generated waves. Such wave-induced
mixing has been studied in tropical cyclone conditions (Ghantous and Babanin, 2014) and
through measurements obtained during tropical cyclones (Toffoli et al., 2012), and this mixing
was shown to cool the surface on a scale of a few hours of cyclone forcing. Turbulence plays
an important role in the heat, momentum, and energy balances of the ocean. Huang et al. (2012)
measured the upper ocean turbulence dissipation associated with wave-turbulence interactions
in the South China Sea. Their results contribute to understanding the SST drop induced by wave
mixing.
The South China Sea (SCS) is one of the largest semienclosed marginal seas subject to frequent
typhoons. Chiang et al. (2011) reported that the average SST cooling in the northern SCS during
typhoon passage was approximately $4.3 \pm 2$ °C in 1958~2008. Tseng et al. (2010) and Lin et al.
(2003) observed an SST drop of more than 9 °C in the northern SCS during Typhoon Kaitak in
2000. They concluded that this drastic SST drop could mainly be ascribed to continual wind-
forced upwelling, a preexisting, relatively shallow thermocline, local bathymetry, and a slow
propagation speed of typhoons. Furthermore, Chiang et al. (2011) estimated that the upwelling
contribution to SST drop is twice that of entrainment for the case of Typhoon Kaitak in 2000.
A larger SST drop in the central SCS was observed by Shang et al. (2008) during Typhoon
Lingling in 2001. Prior to Typhoon Lingling, the SST was approximately 27~30 °C; however,
the SST was reduced by 11 °C after the typhoon passed. This extreme SST drop was mainly
attributed to preexisting eddies that were driven by the northeast monsoon. Zheng et al. (2010)
also considered that preexisting eddy is a favored condition for intensive cooling after typhoon
passage.





SST drops also frequently occur in the waters off northeastern Taiwan. Kuroshio flows through
this region, which is the most important current that transports warm water from the tropical
ocean. The SST drop off northeastern Taiwan mainly occurs during the winter monsoon rather
than the summer season (Tsai et al., 2008a; Jan et al., 2013). Bathymetry-induced upwelling,
rather than entrainment mixing, is considered to be the primary cause of SST drops in this region
(Tsai et al., 2008). The numerical modeling results of Tsai et al. (2008b; 2013) suggest that the
Taiwan Strait outflow is blocked by northerly winds, facilitating Kuroshio intrusion and thus
leading to SST drops during the first half of a typhoon passage. This mechanism is similar to
that involved in the winter monsoon. In contrast, Morimoto et al. (2009) demonstrated that the
northward flow of the Kuroshio is mainly because of the continuous, strong southerly winds,
which accelerate the Kuroshio and force its axis shoreward, resulting in the intrusion of the
Kuroshio towards the shelf and SST drops offshore. Furthermore, Wu et al. (2008) indicated
that internal waves were generated after Typhoon Nari's departure in 2001 and that this was a
minor cause of SST drops. SST drops that occur after typhoon passage are rapid and occur
within a short period of time (Tsai et al., 2013). According to previous studies, these temperature
decreases in the waters off northeastern Taiwan are approximately 4~8 °C after typhoon passage
(Chang et al., 2008; Wu et al., 2008; Tsai et al., 2008a).
Table 1 summarizes the records of SST drops after typhoon passages reported in the literature.
Most studies on drops in SST have been conducted in the open ocean. There have been
comparatively few studies conducted on near-coastal waters (i.e., less than 1 kilometer from the
coastline). In addition, most previous studies on SST drops have been conducted based on
numerical modeling or satellite images because long-term field observations of SSTs are
relatively rare in typhoon-prone areas. Thus, the purpose of this research is to study SST drops
following typhoon passages in coastal areas. Unlike previous studies, this study was conducted
based on an analysis of field data. Coastal SST variability substantially affects both coastal




environmental ecosystems and human activities, and therefore, typhoon-induced coastal SST
variability requires a dedicated study.
**2. Study area and data**
**2.1  Study area**
This research was conducted on the Longdong coast in northeastern Taiwan, as shown in Figure
1. The Longdong coast is characterized by its irregular coastline and rapidly changing
bathymetry. The Longdong coastline is oriented northwest-southeast at approximately 160
degrees from north. The average sea bottom slope at Longdong is ~1/50. An important North
Pacific warm western boundary current, known as Kuroshio, flows along the eastern waters of
Taiwan. The observed maximum flow velocity of Kuroshio varies between 0.7 and 1.4 m/s and
is located at depths ranging from 20 m to 100 m (Jan et al., 2011). The distance between
Taiwan's coast and the main stream of Kuroshio is varied. Morimoto et al. (2009) demonstrated
that the western edge of the Kuroshio stream flows approach Taiwan during the typhoon period.
In this study, the shift in Kuroshio during typhoon Haitang in 2005 is estimated and plotted in
Figure 1, according to Morimoto et al. (2009).
**2.2 Data**
2.2.1 SST measured by moored buoys
SST can be measured by satellite technology, ships, and floating or moored buoys (Matthews,
2013). Satellite observations provide the spatial distribution of SST; however, moored buoys
record the time series of SST. In this study, the main data are the SST recorded by a 2.5-meter
discus-shaped buoy deployed in the water along the Longdong coast. The Longdong buoy was
deployed by the Coastal Ocean Monitoring Center of National Cheng Kung University, as
assigned by the Taiwan Central Weather Bureau (CWB) in 1998. This buoy is approximately



0.6 km off the Longdong coast and is situated in the water at 23 m depth. The buoy is anchored
to the sea bottom. The buoy was equipped with sensors of water and air temperatures, wind,
pressure and wave, as well as power unit, data transmission unit and control unit. Every hour,
the buoy automatically switches on to collect the oceanographic and atmospheric data. The
sampling rates for all sensors are 2 Hz. The sampling duration for wind and wave data is 10
minutes to the hour and it is 1 minute to the hour for pressure and temperature data. The water
temperature sensor is installed at 0.6 m below the sea surface. The procedures of sensor
calibration, system integration, operation and maintenance have been qualified by ISO
9001:1994 since 2000.
The SST is measured by a platinum resistance temperature detector (RTD) which is capable to
cover the range from -10 to 70 degrees Celsius. The sensor provides ±0.1% F.S. accuracy for
critical temperature monitoring applications. Before integrating the temperature sensor with the
buoy, the sensor is submitted to the National Meteorological Instruments Center in CWB for
calibration to confirm the sensor accuracy. All new or retrieved sensors from the field were
submitted for calibration. After integrating the water temperature sensor into the buoy, the
temperature measurements are compared with those of another sensor to confirm the system's
accuracy before sea deployment. The buoy SST data used in this study can be accessed by
https://doi.pangaea.de/10.1594/PANGAEA.895002.
2.2.2 Water temperature measured by tide station
In addition to the Longdong buoy, SST data were also collected from buoys at the Gueishandao,
Suao, and Hualien and tide stations at Linshanbi, Keelung and Fulong, respectively. The
locations of these stations are shown in Figure 1. The buoys at Gueishandao, Suao and Hualien
are 10.0 km, 1.0 km, and 0.6 km from the coast and are situated in the water at depths of 38 m,
20 m, and 21 m, respectively. All tide stations are located inside the harbors and are equipped
with water temperature sensors installed at the bottoms (depth varies from 2 to 5 m) of the




stations. The SST data from tide stations and used in this study can be accessed by
https://doi.pangaea.de/10.1594/PANGAEA.895002.
2.2.3 Current data
Current data observed by acoustic Doppler current profilers (ADCPs) deployed at Longdong
and Linshanbi were also collected and used for validation. The ADCPs were bottom-mounted
and up-looking and measured the current profile of the sea column. Current data from the
Longdong ADCP were collected from June 2008 to June 2009 and data from four typhoons
(Kalmaegi, Fungwong, Sinlaku, and Jangmi) were recorded. The Linshanbi ADCP only
obtained recordings in September 2013, which included data from the passage of Typhoon
Usagi. The current data used in this study can be accessed by
https://doi.pangaea.de/10.1594/PANGAEA.895002.
2.2.4 Satellite images
Except for the field data, multiscale ultra-high resolution (MUR) SST satellite images
(downloaded from the NOAA website:
http://coastwatch.pfeg.noaa.gov/erddap/griddap/jplMURSST.graph?analyzed_sst) were also
collected for cross analysis. In an optimal way, this dataset combines data from the advanced
very high resolution radiometer, moderate imaging spectroradiometer's Terra and Aqua, and
advanced microwave spectroradiometer-EOS instruments to produce 1-km global SST maps.
Data have been released since 2003, and one image is produced per day. The SST images during
Typhoon Jangmi in 2008 were collected in this study.
2.2.5 Spatial wind field
To discuss the possible mechanism of SST drop, the cross-calibrated multiplatform (CCMP)
gridded surface vector winds for the East Asia area (115-130°E, 18-30°N) were collected.
CCMP is one of the productions provided by the scientific research company, Remote Sensing
Systems (RSS), located in California, USA. The CCMP version 2.0 dataset integrates



observations from satellites, moored buoys, and model results and provides a long-term and
high resolution record of global ocean surface (10 m) winds (Wentz et al., 2015). The spatial
and temporal resolutions of CCMP wind are 0.25 degree and 6 hours, respectively. CCMP has
a wide-ranging appeal to users in educational, operational and research environments. In this
study, data obtained during Typhoon Bilis in 2000, Fungwong in 2008, Morakot in 2009 and
Fanapi in 2010 were downloaded from http://www.remss.com/measurements/ccmp.
**2.3 Data quality check**
Checking data quality is necessary and crucial to field data analysis. Incorrect data may yield
misleading results, and inaccurate observations may have a greater negative impact than a lack
of observations. In addition to the satellite image and wind field data that were downloaded
from qualified websites, all field data were strictly verified. The list of field data used in this
study are shown in Table 2. The field measurements are equipped with a solid data quality
checking (QC) system (Doong et al., 2007), including both automatic and manual verifications
of raw data and statistical data, respectively. The automatic machine verification is used to cull
out the suspicious data according to the rationality, continuity, and correlation of data. The
manual verification is used to double check the suspicious data according to spectrum, nearby
observations and the QC engineers' knowledge and experiences. Except for QC procedures,
data are correlated with nearby measurements every month, season and year to develop quality
accuracy (QA) and increase confidence in the data use. Figure 2 shows one SST drop event in
2013 during Typhoon Usagi as an example. The SST drops were measured by the Longdong
buoy, Longdong ADCP, and Linshanbi tide station. The simultaneous observations of SST
drops using different instruments proves that the phenomenon cannot be ascribed to
instrumental error.
**2.4 Typhoons**
There were 108 typhoon datasets observed by the Longdong buoy from 1998 to 2017. Typhoons



are complex atmospheric phenomena and have high variabilities in intensity, moving track, and
speed; therefore, not all typhoons induced SST drops. For forty-three typhoons, significant SST
drops along the coast of Longdong. Table 3 shows the list of the cases. The intensity of the
typhoons is categorized according to the Saffir-Simpson classification method. The maximum
significant wave height of each typhoon is shown in the table. Typhoon parameters are highly
time dependent. The values of typhoon intensity and maximum sustained wind shown in Table
3 are the numbers obtained when the typhoons were closest to Taiwan.
**3.  Data Availability**
The dataset used in this study was deposited in the World Data Center PANGAEA
(https://doi.pangaea.de/10.1594/PANGAEA.895002). The contents and format of the data are
included in the "readme" file provided with the data.
**4. Statistics on coastal SST drop**
**4.1 SST drop determination**
To estimate the scale and rate of each SST drop event, the starting and ending times and
temperatures of an SST drop process were determined. The background SST, which is defined
as the mean SST over the seven days before the SST drop occurrence, is first obtained to
determine the starting point of the event. The starting time of each SST drop event was defined
based on the point at which the water temperature rapidly dropped to a value lower than the
background SST. The lowest SST was the minimum water temperature value during the
typhoon. The ΔSST was the difference between the background SST and the lowest SST. The
duration and further cooling rate of an SST drop event are then estimated. The cooling rate
represents how rapidly a typhoon exerted effects on the ocean.
**4.2 The significant coastal SST drop event**



1. Typhoon Fungwong occurred in 2008 and was a Category II typhoon when it was close to

2. Taiwan. The typhoon exhibited a maximum wind speed of 43 m/s and a minimum central air

3. pressure of 948 hpa. Fungwong occupied an area at 22ºN and 136ºE and traveled approximately

4. along the latitude of 22ºN at an average speed of 4.7 m s$^{-1}$. The intensity of the typhoon

5. increased to that of a medium typhoon during the second half of July 26 and subsequently

6. changed direction to the northwest. Figure 3 shows the track of the typhoon and the time series

7. of the SST, wind speed, wind direction, and significant wave height observed at the Longdong

8. buoy during Fungwong. Before the typhoon approached, the background SST was 29.1 °C. The

9. mean wind speed was lower than 10 m/s, and the wind directions were irregular. On July 28,

10. Fungwong landed on the eastern coast of Taiwan, and the mean wind speed at Longdong rapidly

11. increased and reached a maximum value of 21.4 m s$^{-1}$. The wind direction shifted northward

12. and continued for approximately one day. The significant wave height increased to 7.9 m on

13. July 28 from less than 0.5 m on July 26. Approximately 7 hours later, the SST began to drop.

14. Cold water at a temperature of 16.6 °C was observed on July 29. The total SST drop was 12.5

15. °C within 17 hours. Then, the SST took 35 hours to recover to its background temperature level.

16. Typhoon Fungwong in 2008 induced the maximum SST drop in Longdong.

17. **4.3 Statistical results**

18. To reduce the measurement uncertainty, only SST drops larger than 2 °C were considered in

19. this study. Forty percent (43 of 108) of typhoons triggered a significant SST drop in Longdong

20. in the past 20 years (1998-2017). Among these 43 typhoons, the mean SST drop was 6.1 °C,

21. and the maximum drop was 12.5 °C (Typhoon Fungwong in 2008). The mean drop duration

22. was 24 hours, and the mean recovery duration was 26.1 hours. The mean cooling rate was 0.32

23. °C/hr; however, the maximum cooling rate reached 0.83 °C/hr, which occurred during Typhoon

24. Bilis in 2000. Figure 4 shows the distribution of the SST drop magnitude. Typhoon passages

25. that caused SSTs to drop by 3~4 °C occurred most frequently. Six typhoons caused coastal SSTs



to drop by more than 10 °C. These include Typhoon Bilis in 2000, Fungwong in 2008, Morakot
in 2009, Fanapi in 2010, Matmo in 2014, and Megi in 2016. The typhoon tracks and time series
of SSTs are shown in the Appendix. The intensities of Typhoon Fungwong (Category II) and
Morakot (category I) were relatively weak, but these typhoons induced the largest and second-
largest SST drops on the Longdong coast.
**5. Mechanisms of coastal SST drop**
**5.1 Typhoon dependence**
5.1.1 Typhoon Intensity
The scale of the typhoon-induced SST drop depends on the typhoon's characteristics, such as
the intensity measured by the maximum surface wind speed, moving speed and size. Zhu et al.
(2006) quantified the influence of SST variability on typhoon intensity using a numerical model.
However, this is not the case for the coastal ocean at Longdong. Of the 43 typhoons that
triggered significant coastal SST drops, there were 8 categorized as category I typhoons, 7
category II typhoons, 8 category III typhoons, 8 category IV typhoons, and 8 category V
typhoons. Another 4 typhoons were categorized as tropical storms (TS). The uniform intensity
distribution of all typhoons causing SST drops demonstrates that intensity may not be a
significant factor triggering the coastal SST drop. This can also be validated according to weak
typhoons (for example, Typhoon Hagibis in 2014) that triggered larger coastal SST drops than
stronger typhoons (for example, category IV Typhoon Tembin in 2012). We used both the
minimum central air pressure and central maximum wind speed as typhoon intensity indicators
to understand their influences on SST drops. The regression results show that the determination
coefficients of the typhoon intensity indicators with the SST drop scale ($\Delta$SST) were smaller
than 0.15. Again, it was suggested that typhoon intensity is not the dominant factor that
influences coastal SST drops.



5.1.2 Typhoon track and moving speed
We classify typhoon moving tracks into five paths, as shown in Figure 5. Tracks A, B, and C
represented typhoons that traveled from southeast to northwest. Track A was north of waters
off Longdong, whereas tracks B and C were south of Longdong. Typhoons on track B made
landfall, whereas track C typhoons traveled along Southern Taiwan. The typhoon numbers (of
a total of 43 cases) and their corresponding mean temperature decreases for each track are listed
in Figure 5. Typhoons that traveled along tracks B and C occupied 70% of those typhoons that
triggered SST drops, and the mean decrease in temperature for the sea surface at Longdong is
greater than 6 °C (7.6 °C for track B; 6.4 °C for track C). This indicates that the mean distance
between track C typhoons and Longdong is more than 500 km. Typhoons that traveled along
track A were closer to the waters off Longdong, but of the typhoons that induced an SST
decrease along this track, the scale of SST decrease was relatively small. Typhoons that passed
along the south side of Longdong had greater induced SST drops than other typhoons. These
results were consistent with those of previous studies conducted in the open ocean (Price, 1981;
Wada et al., 2005; 2009), which have proposed that the SST response is larger on the right side
of a typhoon.
Slow-moving typhoons induced larger SST drops in the open sea because they facilitate more
substantial air-sea interactions (Tsai et al., 2008; Wada et al., 2009; Tseng et al., 2010; Kuo et
al., 2011). This study correlated the typhoon moving speeds with the magnitude of coastal SST
drops and found no correlation (coefficient of determination is 0.02).
5.1.3 Typhoon wind distribution
The above results show that the coastal SST drop at Longdong is correlated with the typhoon
track. Therefore, it is interesting to look directly at the wind distribution during typhoons.
Figure 6 shows the CCMP wind patterns for the four significant cases (Typhoon Bilis in 2000;
Fungwong in 2008; Morakot in 2009; and Fanapi in 2010). Because of the output time limitation



for the operational model, the CCMP wind fields are not exactly at the starting time of SST
drop, but the maximum values are different within 2 hours. All 4 cases show strong winds off
the northeast Taiwan waters, and the wind directions are parallel with the Kuroshio direction.
The coverage of the Kuroshio region with large wind speeds is a significant factor. We found
that when the area of strong wind overlapping with Kuroshio is large (for example, Typhoons
Fungwong and Morakot in Figure 6b and 6c), there was a very large SST drop along the
Longdong coast. We suggest that the interaction between typhoon wind and Kuroshio plays an
important role in triggering coastal SST drops in the northeast corner of Taiwan.
**5.2 Vertical Kuroshio intrusion**
Seeking the source of the cold waters is the most interesting issue in this study. Because the
Longdong buoy observation site is located in near-coastal water (0.6 km from the coastline at
23 m water depth), the cold waters may originate from three sources: river discharge from the
land, adjacent surface water, or subsurface water.
The Shuangsi River is the only stream near Longdong. However, the discharge of the river is
small, and the river water temperature ranges between 26 to 30 °C during the summer typhoon
season, although the mean low SST in the waters off Longdong was 21.5 °C. This fact allows
for rejection of the hypothesis that cold waters were supported by land.
We assume that the cold waters were pumping from the subsurface of Longdong. According to
the simultaneous measurement of wind, we observed southerly winds during the SST drop
periods (Figure 3 and Figure 6, as examples). The prevailing wind directions during these
typhoons were between 164° and 189°. The Longdong coastline lies at an angle of 160°
from north. Thus, typhoons created winds parallel to the Longdong coastline and induced
coastal upwelling. The subsurface water is usually cooler than the surface water it replaces. To
prove this assumption, the current profile data were analyzed.
The current profile data were measured very close to the Longdong buoy by an ADCP from



2008 to 2009. There were four typhoon-induced surface cooling cases observed during the
ADCP measurement period: Typhoon Kalmaegi (ΔSST = 5.1 °C), Typhoon Fungwong (ΔSST
= 12.5 °C), Typhoon Sinlaku (ΔSST = 6.8 °C), and Typhoon Jangmi (ΔSST = 8.0 °C). The
current profiles obtained during Typhoon Fungwong are shown in Figure 7. In the waters off
Longdong, currents flowed offshore while the alongshore winds blew during typhoons. The sea
current in the area generally flows shoreward, but instead, the current flowed seaward. The data
demonstrated that typhoons generate an alongshore wind and pump cold water from the
subsurface of Longdong to cool the surface.
The mean SST drop in the waters off Longdong was estimated to be 6.1 °C; however, the
Longdong buoy is situated in water that is 23 m deep. The difference in water temperature
between the sea surface and sea bottom is only approximately 2~3 °C. It was assumed that the
observed cold water was not from the subsurface water at the Longdong buoy location but may
be transferred from offshore deep sea waters. In this study, we referred to the data of the mean
water temperature profile from the Ocean Data Bank (ODB) of the Ministry of Science and
Technology of Taiwan. The data have been collected by research vessels since 1985. At a deep
sea location (122.5°E, 25.25°N) in waters off Longdong, the temperature is 22.9 °C at a depth
of 50 m, 18.8 °C at 100 m and 14.5 °C at 200 m. The mean lowest SST for those 43 events was
21.5 °C and was 16.1 °C for the extreme case. Therefore, we determined that the cold water
was being pumped from a maximum depth of 155 m and then intruded the coastal area. This
finding reaches the maximum value that Narayan et al. (2010) proposed in which cooler waters
from 100-150 m depths are able to be pumped via coastal upwelling.
To identify the movement path of cold water being pumped from the deep ocean, the starting
time of SST drop was assessed at several stations in the research area, as shown in Figure 1.
The analysis results of Typhoon Morakot (ΔSST = 12.3°C) are shown in Table 4 as an example.
The lag time shown in the table is the start time differences in the SST drops between the



stations for the Longdong buoy; in the table, a positive number indicates that the SST drop
observed at the station occurred later than that observed at the Longdong buoy. As Table 4
shows, we found that coastal SST drops occurred earliest in Longdong waters. We suggest,
according to the bathymetry off northeast Taiwan, that the cold waters were pumped from the
Kuroshio subsurface (~155 m depth) in the Okinawa Trough and reached the Longdong area
first, and then, the cold water was transported north to Keelung and south to Suao, respectively.
Figure 8 shows a sketch of the cold water movement path. This assumption can partially prove
that no significant SST drop occurred at the Hualien buoy.
The exchange of water masses off northeastern Taiwan is complex. Chen et al. (1995) showed
that at least six water masses take part in the mixing processes in this region, including the
Kuroshio Surface Water (SW), Kuroshio Tropical Water (TW), Kuroshio Intermediate Water
(IW), East China Sea Water (ECSW), Coastal Water (CW) and the Taiwan Strait Water (TSW).
According to extensive investigations, the intrusion of the Kuroshio into the East China Sea
(ECS) occurs northeast of Taiwan (Hsueh et al., 1992; Tang et al., 1999; Guo et al., 2006; Yang
et al., 2011; Wu et al., 2017; Yang et al., 2018). The mechanism leading to the Kuroshio
intrusion into the ECS is still being researched. Recently, Zhou et al. (2018) indicated that the
Kuroshio subsurface water could intrude into the ECS shelf from northeast Taiwan and reach
north of 29 degrees N. Yang et al. (2018) explained that a topographic beta spiral occurs when
the Kuroshio encounters the shelf break and induces strong upwelling. These researchers
suggested that the topographic beta spiral provides a dynamic channel to bring the cold deep
water from Kuroshio to the continental shelf. Our findings in this study provide direct evidence
from long-term buoy measurements.
**5.3 Spatial cold water intrusion**
In addition to coastal upwelling, the cold water in the coastal area of Longdong may also come
from offshore surfaces, as many studies have confirmed that a cold dome exists in the waters



off northeastern Taiwan. Numerous observational and modeling studies have reported
occurrences of cold water and isotherm doming in northeast Taiwan, which is known as the
cold dome (Tang et al., 1999; Yang et al., 2011; Shen et al., 2011; Jan et al., 2011;
Gopalakrishnan et al., 2013; Cheng et al., 2018). When the Kuroshio flows near the northeastern
Taiwan shelf, a weaker northwestward branch intrudes the ECS shelf (Tang et al., 1999; Lee
and Matsuno, 2007). Recently, Cheng et al. (2008) demonstrated a 4-6 year interannual
variability in the cold dome. Then, the cold dome is formed because of the on-shelf intrusion
of the Kuroshio subsurface water. Gopalakrishnan et al. (2013) established a numerical model
and found that the cold dome occurrences appeared to be connected with the seasonal variability
in the Kuroshio. Jan et al. (2011) used field observation data and satellite images to better
understand that the center of the cold dome is located at approximately 25.625°N, 122.125°E.
The diameter of the cold dome is approximately 100 km, and it has a weak counterclockwise
circulation. The SST of the cold dome is ~ 3°C below the temperature of the ambient shelf
waters.
Daily satellite images (Figure 9) show the spatial distribution of SSTs during Typhoon Jangmi
in 2008. The cold dome moved shoreward along the movement of the typhoons. The
temperature difference between the coastal area of Longdong and the center of the cold water
is generally less than 3 °C. However, the scale of SST drop in the Longdong area was much
higher. Although the contributions from the north (cold dome) and deep sea were not
decomposed, it was suggested that cold water coming from the deep sea dominates the coastal
SST drops in the Longdong area.
**6. Conclusions**
Seawater temperature affects marine environmental ecosystems and human activities. The
variability in seawater temperature also influences typhoon intensity. It is widely known that



the SST may drop a few degrees after passage of a typhoon. However, in this study, we found
that following summer typhoon passages in the coastal waters off Longdong in Taiwan, the SST
may decrease to values lower than the annual minimum temperature (which always occurs in
winter).
Long-term SST field data from the Longdong buoy (which is located 0.6 km offshore at a water
depth of 23 m) over the past 20 years (1998 to 2017) were analyzed to study coastal SST drops.
These decreases were observed after the passage of 43 typhoons. The mean SST drop during
the 43 events was 6.1 °C. The lowest SST was 16.1 °C, which was observed during Typhoon
Morakot in 2009; however, the maximum SST drop was 12.5 °C, observed during Typhoon
Fungwong in 2008. This scale of decrease is much larger than that in the open ocean. The mean
duration of the SST drop was 24 hours, and on average, 26.1 hours were required for the SST
to recover to the background temperature.
Previous studies on the open ocean have proposed that the scale of SST drop is related to
typhoon intensity and speed. However, we found that the scale of typhoon-induced coastal SST
drops in the northeast Taiwan area were mainly correlated with the typhoon track. Typhoon
intensity and moving speed do not appear to be significant factors driving coastal SST drops in
this location. Typhoons that moved south of Longdong (i.e., Longdong is to the right side of the
typhoon) accounted for more than 70% of coastal SST drops and exhibited extremely large
decrease scales, irrespective of whether these typhoons traveled near or far from Longdong.
Wind-driven coastal upwelling was confirmed as the main mechanism involved in substantial
coastal SST drops after typhoon passage at Longdong. The measurements indicated that many
typhoons were accompanied by alongshore winds blowing in a constant direction. Such winds
induce coastal upwelling and pump bottom seawater up to the surface. This was verified
through measurements of the current profile collected at Longdong. This discovery explains the
conclusion that SST drops are mainly influenced by typhoon tracks. However, the cold waters



were not directly supplied from the subsurface of Longdong. We suggest that the coldest water
may originate from depths of 155 m in the Okinawa Trough, which is ~50 km from Longdong.
These waters are the subsurface waters of Kuroshio. We found that the coverage of a large wind
speed region by the Kuroshio is a significant factor that triggers the coastal SST drop. When
the strong wind area largely overlapped with Kuroshio, there was a very large SST drop on the
Longdong coast. By analyzing SST drop processes and the lag times between field stations, we
suggest that the cold water intrudes first at Longdong and is then transported along the coast.
Except for the vertical source of cold water, the cold waters from the known cold dome off
northeastern Taiwan may also penetrate and cool the coastal area. An analysis of satellite images
indicated that the cold dome moves towards the north coast of Taiwan after typhoon passage
and contributes to coastal SST drops. In this study, the contributions of the offshore surface cold
water and Kuroshio subsurface cold water were not decomposed, but we suggest that the
Kuroshio subsurface cold water is the main source of the Longdong coastal SST drop. The
presentation of the coastal SST dataset with significant drop may help to understand the
interaction between Kuroshio with typhoons, and can be used to calibrate and validate the
numerical models of such interactions.
**Author contributions.** D.J. Doong made the main contribution to this paper. He initiated the
idea, collected the data, designed the experiment and wrote the manuscript. J.P. Peng worked
on the data quality check, analysis and plotting the figures. A.V. Babanin joined the discussions
and provided constructive suggestions on writing the manuscript.
**Competing interests.** The authors declare that they have no conflict of interest.
**Acknowledgements**



This research was performed with support from the Ministry of Science and Technology
(MOST) of Taiwan under grant no. MOST 106-2628-E-006-008-MY3. The buoys that measure
SST data are operated by the Coastal Ocean Monitoring Center of National Cheng Kung
University in Tainan, Taiwan. The authors would like to thank all their colleagues at the center.
In addition, the authors acknowledge the Industrial Technology Research Institute (ITRI) for
providing the ADCP current data.
**Appendix: Six cases of coastal SST drops larger greater than 10 °C observed by the**
**Longdong buoy after typhoon passage.** (Left figure shows the typhoon tracks,
and the time series of SSTs are shown on the right.)
**(1) Typhoon Bilis in 2000, max. ΔSST = 10.0 °C**

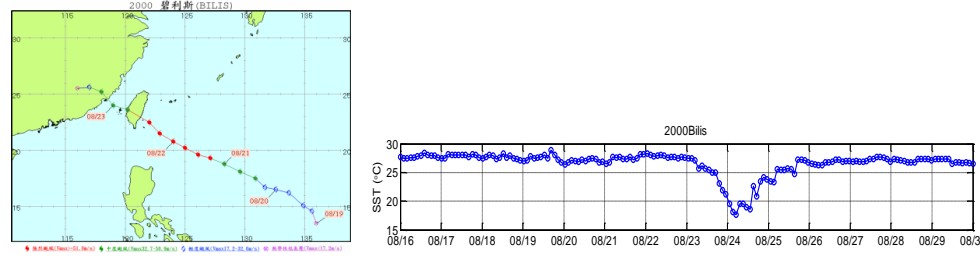

**(2) Typhoon Fungwong in 2008, max. ΔSST = 12.5 °C**

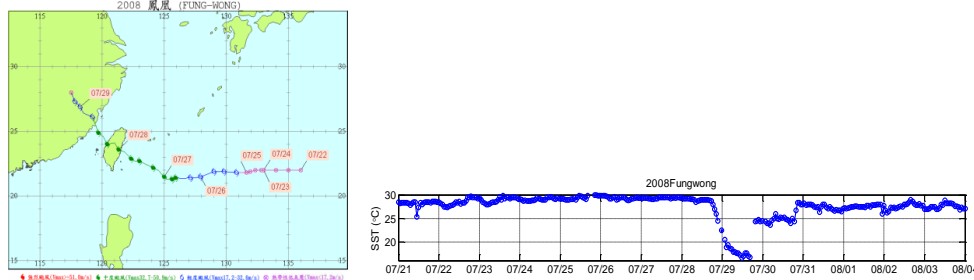

**(3) Typhoon Morakot in 2009, max. ΔSST = 12.3 °C**




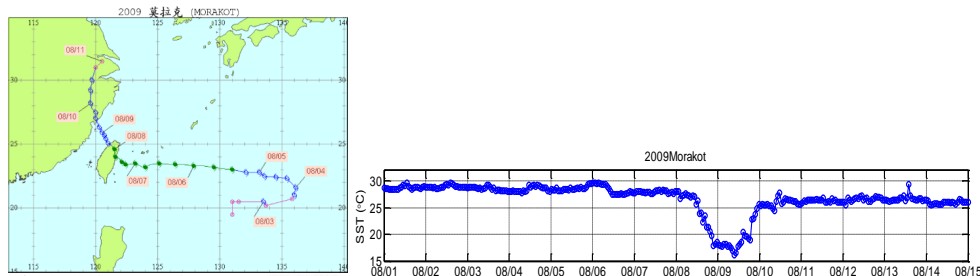
**(4) Typhoon Fanapi in 2010, max. ΔSST = 10.5 °C**
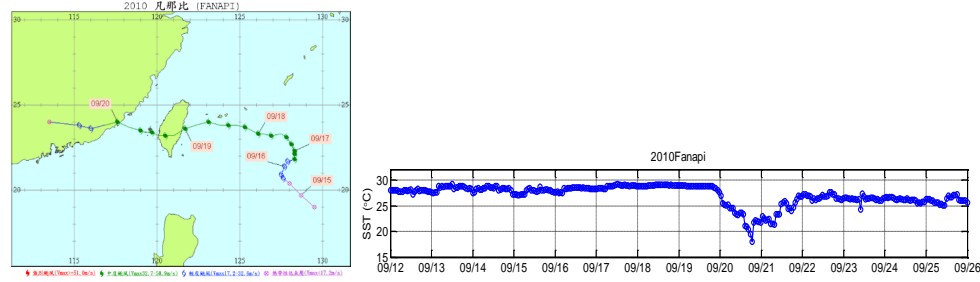
**(5) Typhoon Matmo in 2014, max. ΔSST = 10.4 °C**
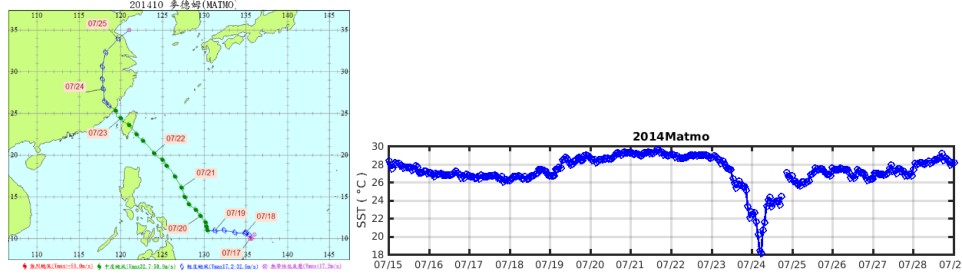
**(6) Typhoon Megi in 2016, max. ΔSST = 10.0 °C**



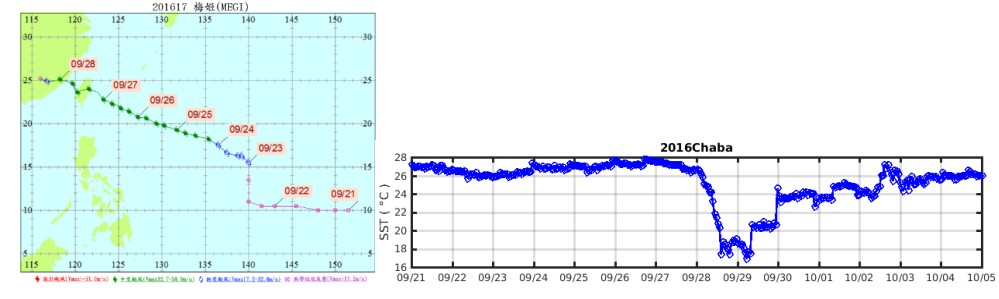





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



**List of Table Captions:**

Table 1 Records of SST drops due to typhoon passage in the literature

Table 2 List of field SST data used in this study

Table 3 Significant SST drops observed at the Longdong buoy (N.E. Taiwan coast) during 43

typhoon passages from 1998 to 2017

Table 4 Quantities of SST drop, the lowest SST and their lag time corresponding to the

Longdong buoy during Typhoon Morakot in 2009. A positive lag time value indicates that the

SST drop observed at the station occurred later than that observed at the Longdong buoy. "-

" means no significant SST drop observed.



1  Table 1 Records of SST drops due to typhoon passage in the literature

| Sea area | SST drop | Typhoon | Main analysis data | Reference |
|---|---|---|---|---|
| Various Regions | 1-8°C | 16 typhoons from 1958 to 1988 | Modeling | Bender et al. (1993) |
| Gulf of Mexico | 2°C | Eloise in 1975 | Field data | Price (1981) |
| N.W. Pacific (off Taiwan coast) | 8°C | Gerald in 1987 | Field data | Tsai et al (2008) |
| N.W. Pacific (off Japan coast) | 9°C | T8914/T8915 in 1989 | Satellite image | Sakaida et al. (1998) |
| SCS | 1°C | Ernie in 1996 | Modeling | Chu et al. (1996) |
| N.W. Pacific (off Taiwan coast) | 9°C | Herb in 1996 | R/V data | Chen et al (2003) |
| India Ocean | 6-7°C | Chennai in 1997 | Modeling | Rao et al (2004) |
| N.W. Pacific | 3°C | Rex in 1998 | Modeling & R/V data | Wada et al (2005; 2009) |
| N. SCS | 9°C | Kaitak in 2000 | Modeling | Tseng et al (2010) |
| N. SCS | 10.8°C | Kaitak in 2000 | Modeling | Chiang et al (2011) |
| M. SCS | 11°C | Lingling in 2001 | Satellite image | Shang et al (2008) |
| N.W. PO (off Taiwan coast) | 5°C | Nari in 2001 | Satellite image | Wu et al (2008) |
| N. SCS | 5.3°C | Krovanh in 2003 | Modeling | Jiang et al (2009) |
| N.W. Pacific (Luzon Strait) | 1.8°C | Dujuan in 2003 | Modeling | Kuo et al (2011) |
| N.E. Pacific (N. Carolina) | 1-3°C | Isabel in 2003 | Field data | Bingham (2007) |
| N.W. Pacific (Kuroshio region) | 3°C | Megi in 2004 | Satellite image | Wei et al. (2014) |
| N.W. Pacific (Kuroshio region) | 4°C | Morakot in 2009 | Modeling and Argo data | Zheng et al. (2014) |
| N.W. Pacific (off Taiwan coast) | 4.5°C | Haitang in 2005 | Satellite image | Chang et al (2008) |
| N.W. Pacific (off Taiwan coast) | 13°C | Haitang in 2005 | Field data | Morimoto et al (2009) |
| N.W. Pacific (Luzon Strait) | 3.5°C | Pabuk in 2007 | Modeling | Kuo et al (2011) |
| N.W. Pacific (off Taiwan coast) | 2-4°C | Fungwong in 2008 | R/V data | Hung et al (2010) |
| SCS | 5-6°C | Nuri in 2008 | Modeling | Sun et al. (2015) |
| N.W. Pacific | 2°C | Kaemi in 2006 | Satellite image | Subrahmanyam (2015) |
| N.W. Pacific | 0.61- | 22 typhoons from | SST maps and | Liu and Wei |





| (Kuroshio region) | 4.93°C | 2001 to 2010 | Argo data | (2015) |
|---|---|---|---|---|
| N.W. Pacific (off Taiwan coast) | 7°C | Morakot in 2009 | Modeling | Tsai et al (2013) |
| SCS | 8°C | Megi in 2010 | Modeling & Satellite image | Ko et al. (2014) |
| SCS | 4.2°C | Megi in 2010 | Modeling & Mooring | Guan et al. (2014) |



1                              Table 2 List of field data used in this study

| Data Type | Instrument Type | Station Name | Location | Depth (m) | Sampling interval (hour) | Accuracy ($^o$C or m/s) |
|---|---|---|---|---|---|---|
| SST | Buoy | Longdong | 121.9219 E; 25.0983 N | 23 | 1 / 2* | 0.1 |
| SST | Buoy | Gueishandao | 121.9233 E; 24.8469 N | 38 | 1 / 2* | 0.1 |
| SST | Buoy | Suao | 121.8800 E; 24.6194 N | 20 | 1 / 2* | 0.1 |
| SST | Buoy | Hualien | 121.6308 E; 24.0356 N | 21 | 1 | 0.1 |
| SST | Tide Station | Keelung | 121.7442 E; 25.1572 N | 5 | 1 | 0.1 |
| SST | Tide Station | Fulong | 121.9500 E; 25.0217 N | 5 | 1 | 0.1 |
| SST and Current | ADCP | Longdong | 121.9219 E; 25.0983 N | 23 | 1 | 0.1 |
| SST | ADCP | Linshanbi | 121.5103 E; 25.2839 N | 24 | 0.1 | 0.1 |

2      * All buoys have sampling interval 2 hours from 1998 to 2003 and 1 hour from 2004 to 2017.





Earth System Science Data Discussions — Open Access

Table 3 Significant SST drops observed at the Longdong buoy (N.E. Taiwan coast) during 43 typhoon passages from 1998 to 2017

| No | Typhoon name | Typhoon dates | Track category* | Intensity category* | Moving speed* (m s⁻¹) | Maximum sustained wind* (m/s) | Max. Hs (m) | ΔSST | Duration of SST drop (hr) | Duration of SST recovery (hr) | Cooling rate (°C/hr) |
|---|---|---|---|---|---|---|---|---|---|---|---|
| 1 | Zeb | 1998/10/10-10/17 | D | V | 6.1 | 38 | 6.0 | 3.7 | 14 | 20 | 0.26 |
| 2 | Babs | 1998/10/14-10/30 | E | IV | 4.2 | 15 | 3.6 | 2.4 | 28 | 4 | 0.09 |
| 3 | Maggie | 1999/6/1-6/9 | C | III | 6.7 | 38 | 4.3 | 5.6 | 14 | 26 | 0.40 |
| 4 | Kaitak | 2000/7/3-7/12 | D | I | 10.6 | 30 | 2.4 | 3.5 | 46 | 50 | 0.08 |
| 5 | Bilis | 2000/8/18-8/27 | B | V | 6.1 | 53 | 5.0 | 10.0 | 12 | 24 | 0.83 |
| 6 | Xangsane | 2000/9/25-10/2 | D | III | 9.2 | 33 | 4.9 | 2.8 | 28 | 92 | 0.10 |
| 7 | Chebi | 2001/6/19-6/24 | E | III | 8.1 | 33 | 2.5 | 3.5 | 24 | 20 | 0.15 |
| 8 | Utor | 2001/7/1-7/7 | C | I | 9.2 | 38 | 5.1 | 9.0 | 24 | 12 | 0.38 |
| 9 | Toraji | 2001/7/25-8/1 | B | III | 4.7 | 38 | 3.2 | 3.6 | 18 | 12 | 0.20 |
| 10 | Nari | 2001/9/5-9/21 | A | III | 1.7 | 40 | 2.3 | 2.6 | 11 | 27 | 0.24 |
| 11 | Lekima | 2001/9/22-9/30 | B | II | 1.4 | 35 | 4.4 | 7.6 | 26 | 38 | 0.29 |
| 12 | Morakot | 2003/7/31-8/4 | B | I | 5.3 | 23 | 1.7 | 4.0 | 28 | 32 | 0.14 |
| 13 | Dujuan | 2003/8/27-9/3 | C | IV | 8.3 | 43 | 5.4 | 6.0 | 12 | 28 | 0.50 |
| 14 | Mindulle | 2004/6/21-7/4 | D | IV | 4.2 | 28 | 3.9 | 8.0 | 15 | 26 | 0.53 |
| 15 | Nockten | 2004/10/14-10/26 | B | III | 5.6 | 40 | 8.2 | 3.5 | 11 | 5 | 0.32 |
| 16 | Matsa | 2005/7/30-8/8 | A | II | 3.9 | 40 | 5.2 | 3.4 | 23 | 14 | 0.15 |
| 17 | Sanvu | 2005/8/9-8/14 | C | I | 6.4 | 20 | 3.2 | 7.3 | 23 | 6 | 0.32 |
| 18 | Longwang | 2005/9/25-10/3 | B | IV | 6.4 | 51 | 7.5 | 6.7 | 22 | 14 | 0.30 |
| 19 | Chanchu | 2006/5/8-5/18 | E | IV | 11.9 | 25 | 2.7 | 3.5 | 24 | 6 | 0.15 |
| 20 | Bilis | 2006/7/8-7/16 | B | TS | 5.0 | 25 | 4.8 | 5.3 | 12 | 30 | 0.44 |



| | | | | | | | | | | | |
|---|---|---|---|---|---|---|---|---|---|---|---|
| 21 | Kaemi | 2006/7/17-7/27 | B | I | 4.7 | 38 | 3.3 | 7.6 | 30 | 47 | 0.25 |
| 22 | Sepat | 2007/8/12-8/20 | B | V | 5.6 | 48 | 4.8 | 9.5 | 30 | 8 | 0.32 |
| 23 | Kalmaegi | 2008/7/13-7/20 | B | II | 5.6 | 33 | 3.1 | 5.1 | 15 | 18 | 0.34 |
| 24 | Fungwong | 2008/7/23-7/30 | B | II | 4.7 | 43 | 7.9 | 12.5 | 17 | 35 | 0.74 |
| 25 | Sinlaku | 2008/9/8-9/21 | B | IV | 2.2 | 38 | 7.3 | 6.8 | 18 | 20 | 0.38 |
| 26 | Jangmi | 2008/9/23-10/1 | B | V | 5.0 | 51 | 11.2 | 8.0 | 19 | 44 | 0.42 |
| 27 | Morakot | 2009/8/2-8/11 | B | I | 3.3 | 35 | 8.2 | 12.3 | 20 | 16 | 0.62 |
| 28 | Merant | 2010/9/6-9/10 | C | I | 3.3 | 15 | 1.5 | 4.6 | 26 | 42 | 0.18 |
| 29 | Fanapi | 2010/9/14-9/21 | B | III | 5.6 | 45 | 7.2 | 10.5 | 21 | 26 | 0.50 |
| 30 | Nanmadol | 2011/8/21-8/31 | C | V | 2.5 | 35 | 2.9 | 8.9 | 27 | 30 | 0.33 |
| 31 | Saola | 2012/7/26-8/5 | B | II | 4.2 | 30 | 8.3 | 5.4 | 10 | 14 | 0.54 |
| 32 | Tembin | 2012/8/17-8/30 | D | IV | 3.1 | 30 | 2.5 | 3.8 | 10 | 42 | 0.38 |
| 33 | Trami | 2013/8/16-8/24 | A | I | 12.8 | 30 | 3.1 | 2.4 | 21 | 10 | 0.11 |
| 34 | Usagi | 2013/9/16-9/24 | C | V | 5.3 | 53 | 4.3 | 6.4 | 20 | 41 | 0.32 |
| 35 | Hagibis | 2014/6/13-6/18 | E | TS | 3.6 | 15 | 1.0 | 4.5 | 76 | 60 | 0.06 |
| 36 | Matmo | 2014/7/16-7/25 | B | II | 5.6 | 38 | 4.3 | 10.4 | 22 | 29 | 0.47 |
| 37 | Fungwong | 2014/9/17-9/24 | D | TS | 6.1 | 25 | 3.4 | 3.5 | 43 | 10 | 0.08 |
| 38 | Nepartak | 2016/7/2-7/10 | B | V | 4.7 | 55 | 3.6 | 7.5 | 36 | 29 | 0.21 |
| 39 | Meranti | 2016/9/8-9/16 | C | V | 5.6 | 58 | 3.9 | 8.3 | 21 | 19 | 0.40 |
| 40 | Megi | 2016/9/22-9/29 | B | IV | 6.4 | 45 | 12.5 | 10.0 | 29 | 18 | 0.34 |
| 41 | Aere | 2016/10/4-10/14 | C | TS | 6.4 | 18 | 3.9 | 2.6 | 42 | 46 | 0.06 |
| 42 | Nesat | 2017/7/25-7/30 | B | II | 4.2 | 40 | 2.4 | 6.3 | 11 | 22 | 0.57 |
| 43 | Hato | 2017/8/19-8/24 | C | III | 7.8 | 20 | 2.0 | 5.0 | 51 | 9 | 0.10 |

* indicates that the values were obtained when typhoons were close to Taiwan.



Table 4 Quantities of SST drop, the lowest SST and their lag time corresponding to
the Longdong buoy during Typhoon Morakot in 2009. A positive lag time value
indicates that the SST drop observed at the station occurred later than that observed at
the Longdong buoy. "-" means no significant SST drop observed.

| SST Station | Lowest SST (°C) | ΔSST (°C) | Lag time (hr) |
|---|---|---|---|
| Linshanbi | 27.0 | < 2°C | - |
| Keelung | 24.7 | 2.6 | +10 |
| Longdong | 16.1 | 12.3 | 0 |
| Fulong | 20.7 | 7.8 | +1 |
| Guishandao | 19.9 | 8.1 | +3 |
| Suao | 17.9 | 11.4 | +6 |
| Hualien | | < 2°C | - |



**List of Figure Captions:**
Figure 1 Locations of the study area and field stations. The gray belt is the main
stream of Kuroshio; however, the dashed gray belt is the shift of Kuroshio during
Typhoon Haitang in 2005 according to measurements by Morimoto et al. (2009)
Figure 2 SST drop observed by various types of instruments during Typhoon Usagi in

6   2013

Figure 3 The significant SST drop event after the passage of Typhoon Fungwong in
2008. (a) The typhoon track; (b) SST; (c) wind speed and direction; and (d)
significant wave height. The data were observed by a data buoy in the Longdong
coastal waters of northeast Taiwan.
Figure 4 Distribution of the SST drop magnitude for 43 typhoons
Figure 5 The SST drops for various typhoon tracks. The two numbers in parentheses
show the typhoon number and the mean SST drop magnitude in the corresponding
typhoon track.
Figure 6 Wind patterns at the time close to the start of the SST drop. (a) Typhoon Bilis
in 2000. The SST started to decrease on 2000/8/23 at 10:00. The wind pattern was
observed on 2000/8/23 at 08:00. (b) Typhoon Fungwong in 2008. The SST started to
decrease on 2008/7/28 at 18:00. The wind pattern was observed on 2008/7/28 at
20:00. (c) Typhoon Morakot in 2009. The SST started to decrease on 2009/8/8 at



13:00. The wind pattern was observed on 2009/8/8 at 14:00. (d) Typhoon Fanapi in
2010. The SST started to decrease on 2010/9/19 at 22:00. The wind pattern was
observed on 2010/9/19 at 20:00
Figure 7 Current profile and corresponding tide level observed in Longdong during
Typhoon Fungwong in 2008
Figure 8 The suggested movement path of cold water. The cold water was pumped
from the Kuroshio subsurface in the Okinawa Trough and reached the Longdong
coastal waters first. Then, the cold water was transported north to Keelung and south
to Suao.
Figure 9 Movement of the cold dome off northeast Taiwan during Typhoon Jangmi in
2008. The typhoon track is shown in the upper panel. The lower panel shows the
satellite images of SST.

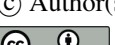



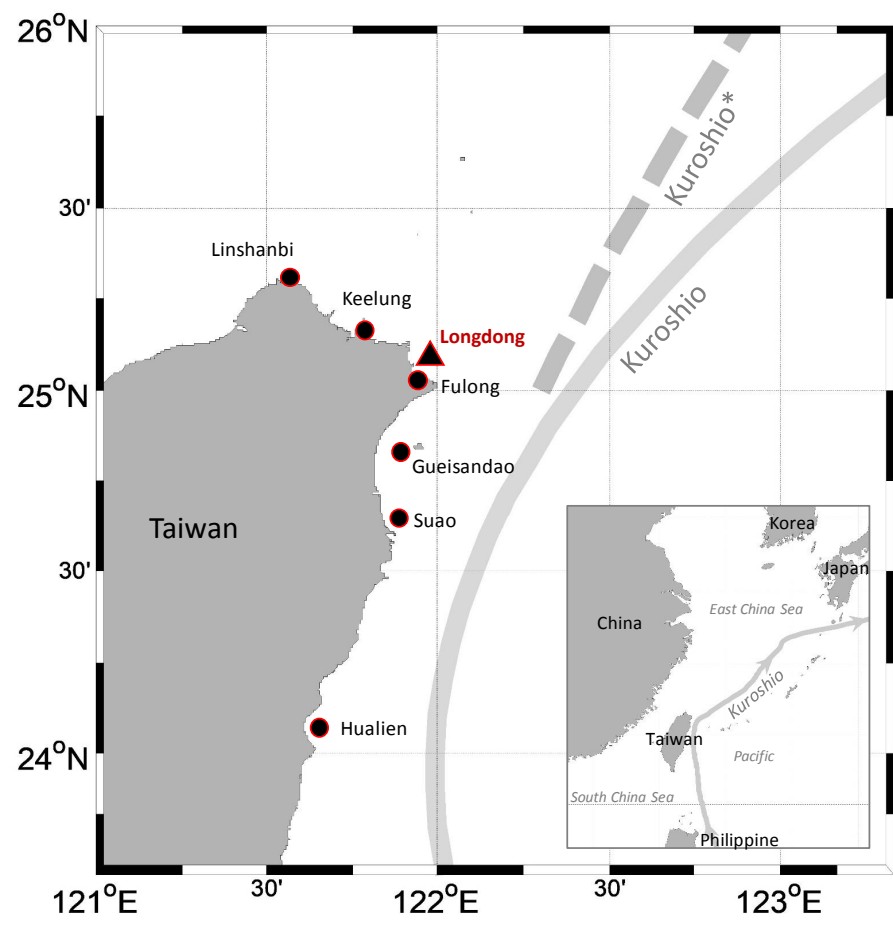

2      Figure 1 Locations of the study area and field stations. The gray belt is the main

3      stream of Kuroshio; however, the dashed gray belt is the shift of Kuroshio during

4      Typhoon Haitang in 2005 according to measurements by Morimoto et al. (2009)



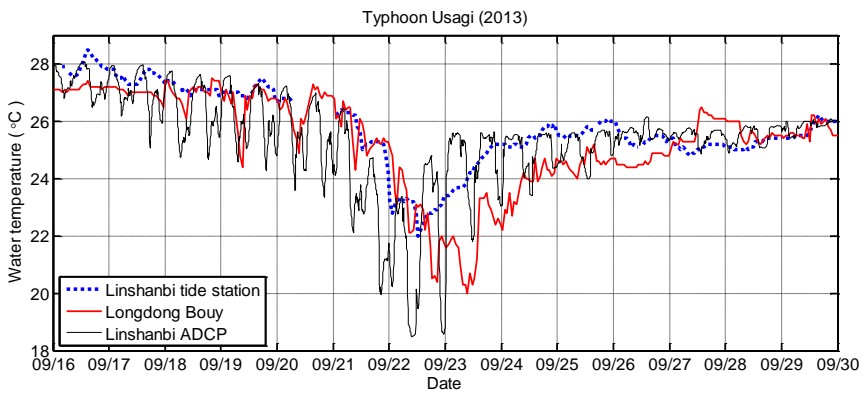

2     Figure 2 SST drop observed by various types of instruments during Typhoon Usagi in

3                          2013



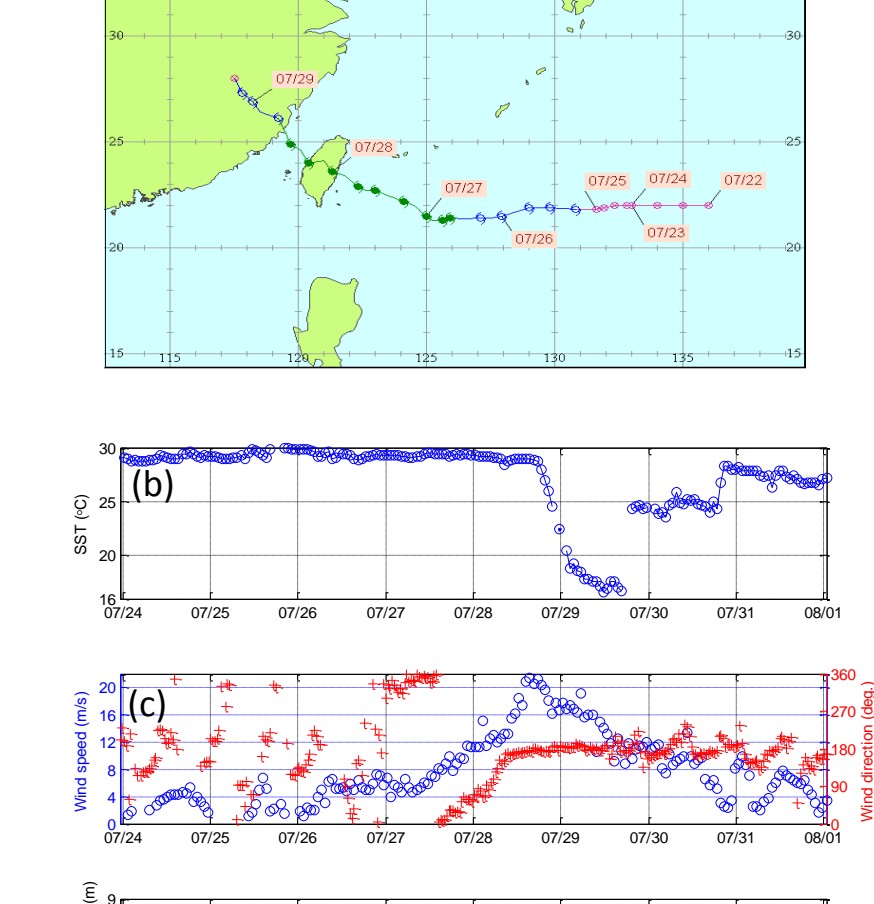

3    Figure 3 The significant SST drop event after the passage of Typhoon Fungwong in

4          2008. (a) The typhoon track; (b) SST; (c) wind speed and direction; and (d)

5      significant wave height. The data were observed by a data buoy in the Longdong

6                      coastal waters of northeast Taiwan.



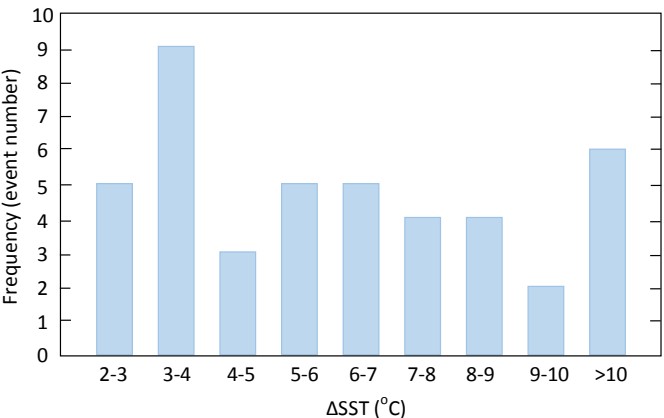

2                      Figure 4 Distribution of the SST drop magnitude for 43 typhoons

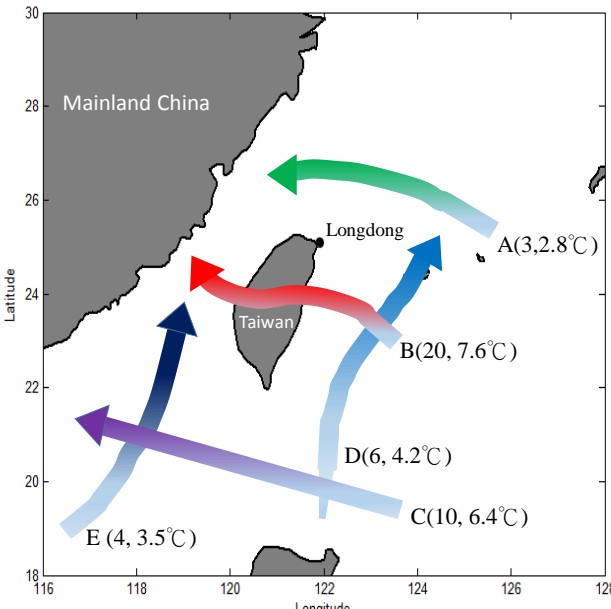

4        Figure 5 The SST drops for various typhoon tracks. The two numbers in parentheses

5         show the typhoon number and the mean SST drop magnitude in the corresponding

6                                    typhoon track.



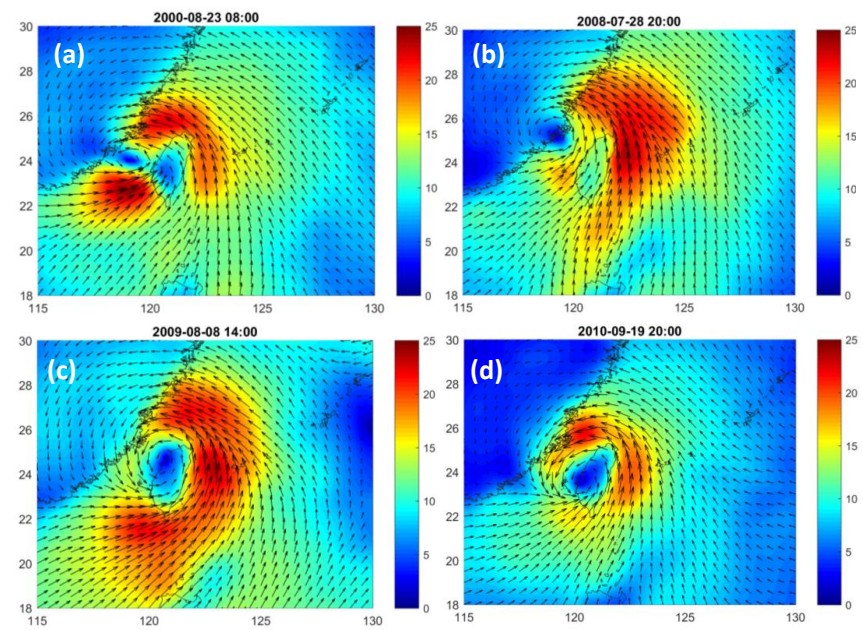

Figure 6 Wind patterns at the time close to the start of the SST drop. (a) Typhoon Bilis in 2000. The SST started to decrease on 2000/8/23 at 10:00. The wind pattern was observed on 2000/8/23 at 08:00. (b) Typhoon Fungwong in 2008. The SST started to decrease on 2008/7/28 at 18:00. The wind pattern was observed on 2008/7/28 at 20:00. (c) Typhoon Morakot in 2009. The SST started to decrease on 2009/8/8 at 13:00. The wind pattern was observed on 2009/8/8 at 14:00. (d) Typhoon Fanapi in 2010. The SST started to decrease on 2010/9/19 at 22:00. The wind pattern was observed on 2010/9/19 at 20:00





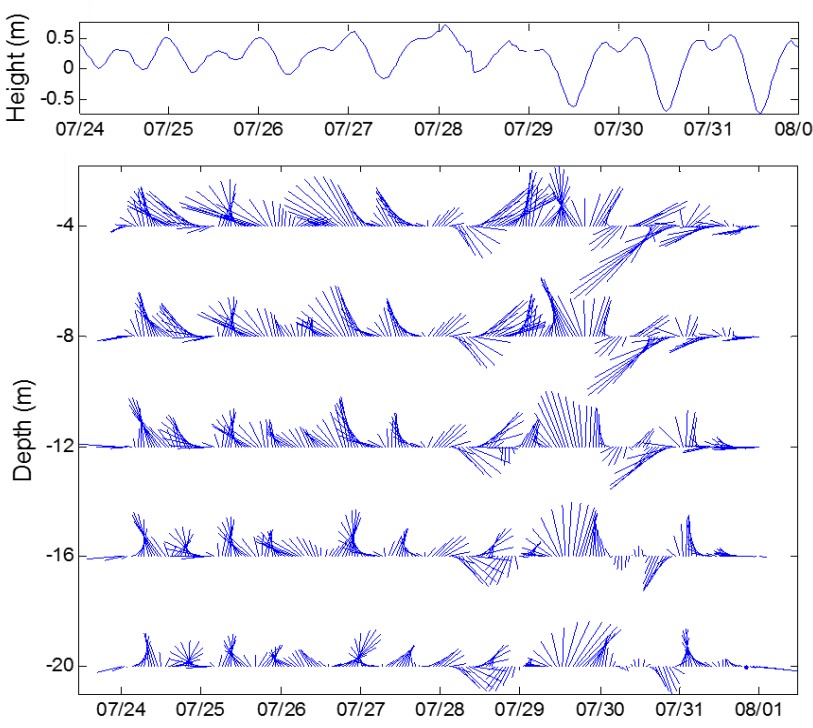

2     Figure 7 Current profile and corresponding tide level observed in Longdong during

3                           Typhoon Fungwong in 2008





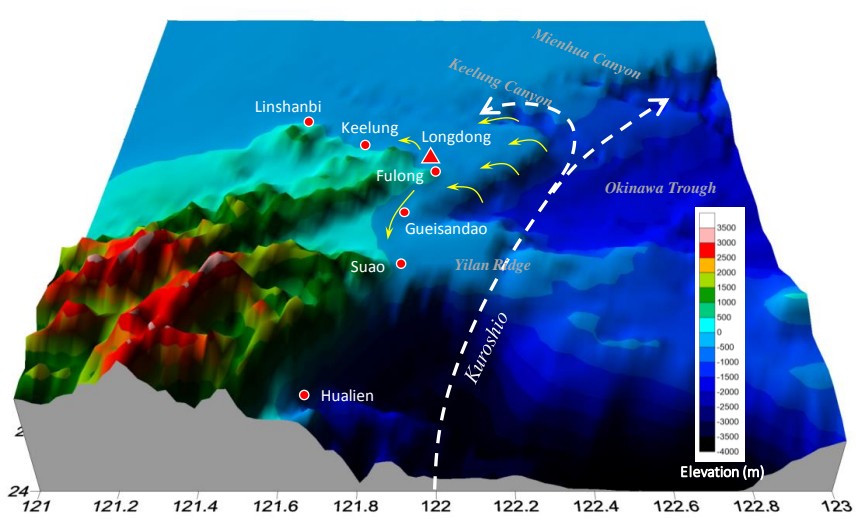

Figure 8 The suggested movement path of cold water. The cold water was pumped
from the Kuroshio subsurface in the Okinawa Trough and reached the Longdong
coastal waters first. Then, the cold water was transported north to Keelung and south
to Suao.



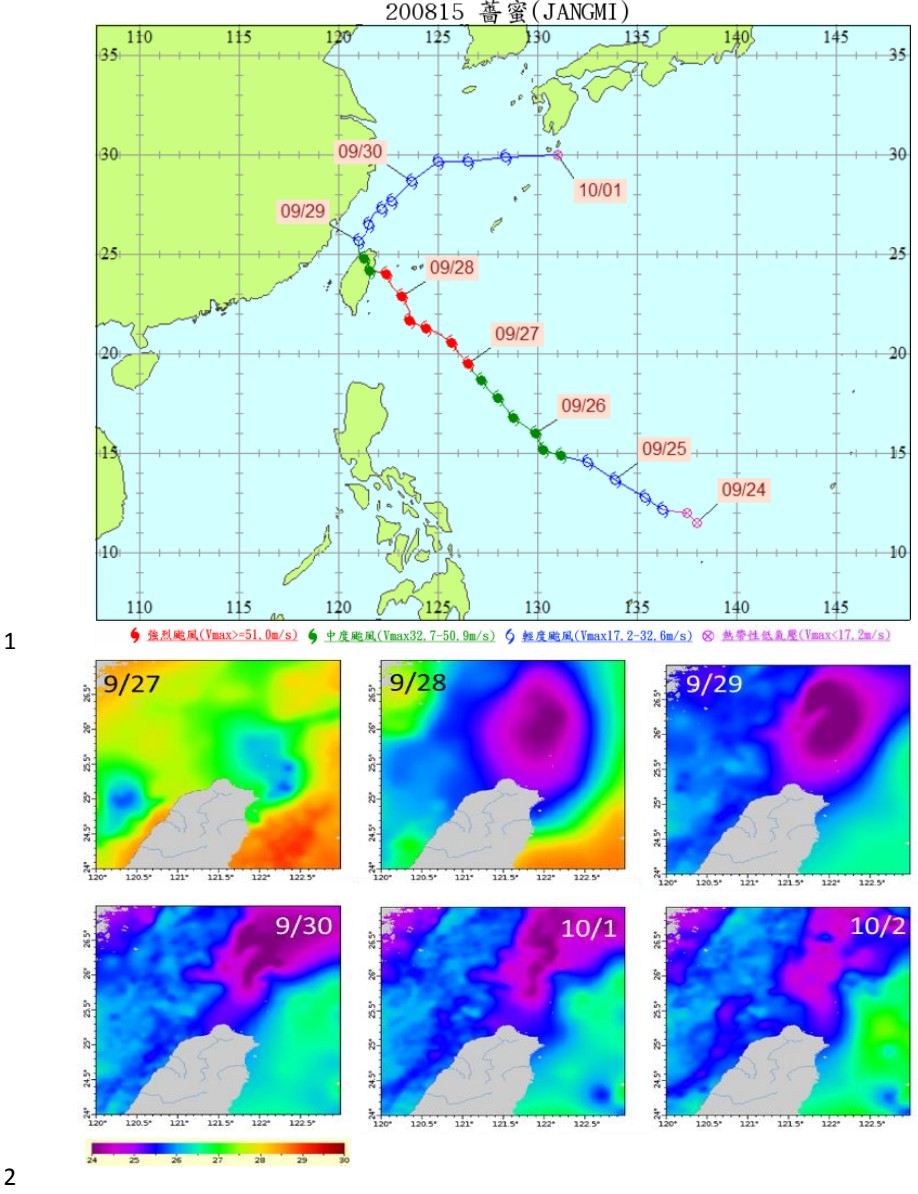

3    Figure 9 Movement of the cold dome off northeast Taiwan during Typhoon Jangmi in

4        2008. The typhoon track is shown in the upper panel. The lower panel shows the

5                    satellite images of SST.