# Peer review of "Field Investigations of Coastal Sea Surface Temperature Drop"

_Earth System Science Data, 2018_

## Referee Comment (RC1) · Anonymous Referee #1 · 15 Nov 2018

Field observations of sea surface temperature (SST) drops are discussed after the passage of several typhoons off Taiwan. Whereas temperature drops have been observed and discussed at offshore locations, this manuscript focuses on a near shore measurement station, where currents and river discharge complicate the physical processes. The authors state that the purpose of their research is to study the sea surface temperature drops with the specific intent to understand the possible mechanisms (see line 6 in the abstract). The data analysis presented in the manuscript, however, is superficial and inconclusive. There is no effort to perform a detailed analysis with e.g. standard regression techniques that can underpin correlation between the SST drops and other environmental conditions. In my view, the authors only speculate on possible mechanisms without presenting robust and sufficiently clear evidence to explain mech-

anisms responsible for the observed sea surface temperature anomalies. Overall, I do not think this manuscript is suitable for publication.

Specific comments:

1. In the abstract, the authors states that an extensive data analysis is presented. However, this extensive analysis is not supported by any of the figures in the manuscript.

2. The authors collected a huge and comprehensive data set of sea surface temperature, wave parameters, atmospheric conditions, etc... and yet there is no attempt to correlate the sea surface temperature with any other environmental variables.

3. ADCP's seem to collects sea surface temperature too as shown in figure 2. However, measuring SST is not mentioned as a capability of the ADCP in section 2.2.3.

4. Satellite images were collected for this study, but it seems they were analysed.

5. Section 2.3 on quality control is potentially interesting. However, the description of quality check is too general. What data did the authors remove? Why? What were the criteria or thresholds for quality control? Much more details are required for the reader to understand the procedure.

6. Figure 2 shows the comparison of SST time series at different instrumentations and it is used to claim that the drops are a consistent feature. However, I do not see how figure 2 can fit the quality check section.

7. There is an unnecessary repetition of data being archived on pangaea.

8. Section 4 on statistics of SST drops is quite misleading. About 2/3 of the section is not related to statistical properties. The remaining part (section 4.3) is just a simplistic description of average values, and it is far from being a rigorous statistical analysis

9. Section 5 is too general and inconclusive. There is an attempt to correlate SST drops with typhoon characteristics such as intensity, but this is done in relation to typhoon categories only. It would be much more meaningful to present scatter plots
of SST drops against environmental parameters, such as wind speed, wave height, pressure, etc.. and then perform machine learning or regression analysis to find correlations, trends etc. The same comments apply for sections 5.1.1, 5.1.2 and 5.1.3. In the present form, these subsections are supported by inconclusive figures that only allow authors to speculate on possible causes, without suggesting a feasible and well supported explanation for SST drops.

10. Another example of speculative discussion is the one in section 5.2. The authors analyse the influence of ocean current on the SST drops, but the only supporting figures are figure 7 and 8. How can a reader infer a correlation between the observed SST drops and current from a plot of bathymetry with overlaid arrows or current directions?

11. Overall, I feel the manuscript misses the extensive data analysis promised in the abstract, making the discussion speculative and the manuscript inconclusive.

---

## Referee Comment (RC2) · Anonymous Referee #2 · 28 Nov 2018

Field investigations of coastal sea surface temperature (SST) drop after typhoon passages is discussed passage at different coastal buoy stations off Taiwan. temperature drops have been observed and discussed at offshore locations. The authors state that the purpose of their research is to study the sea surface temperature drops with the specific aim to understand the possible mechanisms. The data analysis presented in the manuscript, however, is superficial and inconclusive. the comments for this paper as follows: 1. from the moored buoys, we can obtain the profile data even though the depth is 23 m, it is possible to keep more instruments to obtain the data. if so, why authors do not use the different depths data to explain the mechanism? 2. authors used current data but did not mention in which depths the data has been used. 3. authors mentioned that the relation between SST drop and intensity of the typhoon, however,

they did not give any relation between them. 4. from the figure 8, authors mentioned that Kuroshio waters intruded to long dong and other observation areas, however at long dong buoy area, the depth is only 23 m, how the Kuroshio waters come to the surface of the long dong buoy surface? authors need to check properly and if it is so, need to clear explanation about it.

overall, the authors used many data sets, but the interpretation and discussion are not inconclusive.

---

## Author Comment (AC1) · 15 Jan 2019

**Reply to Anonymous Referee #1**

**Overall Comments:**

The reviewer thought the data analysis presented in the manuscript is superficial and inconclusive and had no effort to perform a detailed analysis with e.g. standard regression techniques that can underpin correlation between the SST drops and other environmental conditions. The reviewer thought this manuscript only speculate on possible mechanisms without presenting robust and sufficiently clear evidence to explain mechanisms responsible for the observed sea surface temperature anomalies.

**Overall reply:**

The aim of the journal Earth System Science Data (ESSD) that this manuscript submitted is to publish the articles on original research data (sets), furthering the reuse of high-quality data of benefit to Earth system sciences. We think our manuscript have satisfied with this aim.

We have published the coastal SST data (and corresponding data used in this manuscript) on the Data Center PANGAEA which suggested by ESSD and shown the link DOI in the manuscript which ESSD asked. The coastal SST drop data are rare and valuable especially under the typhoon conditions. This phenomena is found near the coast (0.6 km to coastline) at the Kuroshio edge. Kuroshio cold water intrusion to the continental shelf was recently reported (Zhou et al., 2018; Yang et al., 2018). The dataset presented in this manuscript can provide more evidences (from field measurements) to have better understandings on the mechanisms.

We don't fully agree with the reviewer's comments on that the manuscript is superficial and inconclusive. For data quality, we have shown the instrumentations, sampling methods, sensor calibration, and the data quality check methods (Sections 2.2 and 2.3). Maybe the data quality processes was not shown sufficiently in the original manuscript, we have added more detail description. The data used in this study (from marine data buoy) was cross-compared with various types of instruments (Figure 2) to show more confidence on the data obtained. The reviewer mentioned there is no effort to perform a detailed analysis with e.g. standard regression techniques that can underpin correlation between the SST drops and other environmental conditions. Actually we have done. The results are shown in line 23-24 of page 12 and line 19-20 of page 13 in the original manuscript. We thought to publish the datasets is more crucial for ESSD therefore we didn't show the figures. Now we add them to the revised manuscript. Please see in Figure 5(a)(b) and Figure 7.

This manuscript reports the SST drop phenomena from field measurement. We collect and analyze extensive datasets (from buoys, tide stations, ADCPs and satellite images) and try figure out the potential mechanism of coastal SST drop. We think this manuscript contributes to proof that the Kuroshio cold water intrusion to the continental shelf.

We have revised the manuscript and marked the revisions by blue color.

**Specific comments from the reviewer 1 and the replies:**

1. In the abstract, the authors states that an extensive data analysis is presented. However, this extensive analysis is not supported by any of the figures in the manuscript.

**Reply:**

The conclusions presented in this study (significant SST drop at the coastal ocean after typhoon passage occurs and it is assumed due to the Kuroshio subsurface cold water intrusion to the continental shelf) depends on the qualified field data and their statistics (Section 2.2, 2.3 and Section 4), correlation and discussions with the typhoon parameters (Section 5.1.1 and 5.1.2) and the proofs of current profile (Sec 5.1 & Figure 9) and drop time lag (Table 4). The regression results which originally didn't shown in the manuscript are now added. Please find in Figure 5(a)(b) and Figure 7.

2. The authors collected a huge and comprehensive data set of sea surface temperature, wave parameters, atmospheric conditions, etc... and yet there is no attempt to correlate the sea surface temperature with any other environmental variables.

**Reply:**

In section 5.1, we discussed about typhoon dependence with the respect of coastal SST drop, we learned qualitatively the coastal SST drop is not related to typhoon central air pressure and typhoon central wind speed as well as typhoon moving speed, but typhoon track. Actually we also did the quantitative regression with typhoon parameters and have shown the coefficients of determination in the original submitted manuscript. In the revised manuscript, we show the figures. Please see Figure 5(a)(b) and Figure 7.

This result partially confirmed that the coastal SST drop is not highly related

environmental variables, leading us not to invest time on correlating coastal SST drop with every environmental variables; instead, putting more efforts on where and how the cold water intrude. This result also leading us to believe that those environmental variables can't induce such a significant drop in SST, unless there is a cold water intrusion. That's why we focused more on where and how the cold water intrude in this study as written in section 5.2 and 5.3.

3. ADCP's seem to collects sea surface temperature too as shown in figure 2. However, measuring SST is not mentioned as a capability of the ADCP in section 2.2.3.

**Reply:**

We added a description in the section 2.2.2 that ACDP is capable to measure water temperature and also being collected in this study. Since 2.2.3 is describing about current data, section 2.2.2 is more suitable for describing anything related water temperature data.

4. Satellite images were collected for this study, but it seems they were analysed.

**Reply:**

Yes, they are one-day averaged SST images. They are from the advanced very high resolution radiometer, moderate imaging spectroradiometer's Terra and Aqua, and advanced microwave spectroradiometer-EOS instruments to produce the 1-km global SST maps, done by NOAA.

5. Section 2.3 on quality control is potentially interesting. However, the description of quality check is too general. What data did the authors remove? Why? What were the criteria or thresholds for quality control? Much more details are required for the reader to understand the procedure.

**Reply:**

More description have added in Section 2.3. For detail of the meteo-oceanographic data quality check, please refer to Doong et al. (2007).

Doong, D. J., Chen, S. H., Kao, C. C., and Lee, B. C.: Data quality check procedures of an operational coastal ocean monitoring network, Ocean Eng., 34, 234-246, https://doi.org/10.1016/j.oceaneng.2006.01.011, 2007.

6. Figure 2 shows the comparison of SST time series at different instrumentations and it is used to claim that the drops are a consistent feature. However, I do not see how figure 2 can fit the quality check section.

**Reply:**

Figure 2 show the simultaneous observations of SST drop obtained from 3 different kinds of instrument, illustrating the significant coastal SST drop is not due to instrumental error but a nature phenomenon. All the data from 3 different instrumentations satisfied with the quality check criteria. They are reasonable data (within instrument measurement range and in reasonable environmental range) and the time series is in continuous changes.

7. There is an unnecessary repetition of data being archived on pangaea.

**Reply:**

This is necessary to show the DOI of the dataset in the manuscript, asked by ESSD.

8. Section 4 on statistics of SST drops is quite misleading. About 2/3 of the section is not related to statistical properties. The remaining part (section 4.3) is just a simplistic description of average values, and it is far from being a rigorous statistical analysis

**Reply:**

We planned to show the statistics of coastal SST drops, for example the mean and max. SST drop, the cooling rate, and the drop and recovery durations. We think the numbers will be very high interested by physical oceanographers. To find the statistics, it is necessary to determinate the correct starting and ending drop time which shown in Section 4.1. We think the contents and results are sufficient to express our findings. If possible, we hope the reviewer to suggest clearly the contents of "rigorous statistical analysis".

9. Section 5 is too general and inconclusive. There is an attempt to correlate SST drops with typhoon characteristics such as intensity, but this is done in relation to typhoon categories only. It would be much more meaningful to present scatter

plots of SST drops against environmental parameters, such as wind speed, wave height, pressure, etc.. and then perform machine learning or regression analysis to find correlations, trends etc. The same comments apply for sections 5.1.1, 5.1.2 and 5.1.3. In the present form, these subsections are supported by inconclusive figures that only allow authors to speculate on possible causes, without suggesting a feasible and well supported explanation for SST drops.

**Reply:**

Please refer to our replies for comment 2.

We would like to emphasize that we focused more on detecting where and how the cold water intrusion. In section 5.2 and 5.3, we concluded that typhoon induced southward alongshore winds generating Ekman transport and Kuroshio water intrusion, resulting a massive coastal SST drop. This mechanism is confirmed by field data in figure 6 and 7.

10. Another example of speculative discussion is the one in section 5.2. The authors analyse the influence of ocean current on the SST drops, but the only supporting figures are figure 7 and 8. How can a reader infer a correlation between the observed SST drops and current from a plot of bathymetry with overlaid arrows or current directions?

**Reply:**

Note: Figure 7 and 8 in original manuscript are Figure 9 and 10 in the revised manuscript.

The whole story of potential mechanism of coastal SST drop starts from wind data (in Figure 8) with current profile data afterwards (Figure 9). These two figures confirmed that coastal upwelling is the potential mechanism of significant coastal SST drop. The time lag analysis of SST drops in several stations (shown in Table 4) provides preliminary result of the propagation path of the cold water. Figure 10 was plotted to suggest the movement path of cold water.

11. Overall, I feel the manuscript misses the extensive data analysis promised in the abstract, making the discussion speculative and the manuscript inconclusive.

**Reply:**

Thanks for the reviewer's comments.

We accept part of the reviewer's comments and have made revisions or added new figures, but we don't fully agree with the overall assessment. We think the manuscript satisfies with ESSD's goal and will contribute to have more understanding on the Kuroshio's intrusion to the continental shelf. Since the study area is near Taiwan but similar phenomenon may occur in the other coastal areas.

---

## Author Comment (AC2) · 15 Jan 2019

**Reply to Anonymous Referee #2**

Comments:

1. from the moored buoys, we can obtain the profile data even though the depth is 23 m, it is possible to keep more instruments to obtain the data. if so, why authors do not use the different depths data to explain the mechanism?

**Reply:**

There are various types of ocean buoy. The data buoy we used in this study is for meteo-oceanographic elements measurements, such as wind, wave and temperature. It is like US NOAA/NDBC's buoy (https://www.ndbc.noaa.gov/) or commercial buoys like Fugro SEAWATCH buoy (https://www.fugro.com/about-fugro/our-expertise/technology/seawatch-metocean-buoys-and-sensors). This type buoy has very loose mooring line to maintain a good wave following capability. Therefore the sea temperature sensor is normally mounted at buoy. Only sea surface temperature (~1.5m) is obtained. It is not like TAO buoys which measure the temperature profile. Only near sea surface temperature is measured and analyzed.

2. authors used current data but did not mention in which depths the data has been used.

**Reply:**

We've added this information in the Section of 2.2.3 current data. The current profile data measured by ADCPs are from water depth –4 to –23 m (Note: current profile data only from ADCPs, not from buoys). The current profile data are used in Section 5.2 and Figure 7 to demonstrate that typhoons pump cold water from the subsurface of Longdong to cool the surface.

3. authors mentioned that the relation between SST drop and intensity of the typhoon, however, they did not give any relation between them.

**Reply:**

In section 5.1, we discussed about typhoon dependence with the respect of coastal SST drop, we learned qualitatively the coastal SST drop is not related to typhoon central air pressure and typhoon central wind speed as well as typhoon moving speed, but typhoon track. Actually we also did the quantitative regression with typhoon

parameters and have shown the coefficients of determination in the original submitted manuscript. In the revised manuscript, we show the figures. Please see Figure 5(a)(b) and Figure 7.

4. from the figure 8, authors mentioned that Kuroshio waters intruded to Longdong and other observation areas, however at Longdong buoy area, the depth is only 23 m, how the Kuroshio waters come to the surface of the long dong buoy surface?

Reply:

This study suggested the Kuroshio subsurface cold waters intruded to Longdong (locates at 0.6 m depth and 0.6 km to coastline) according to the measured data and the analysis results. But how the propagation of the cold water is not in the scope of this research. Zhou et al. (2018) and Yang et al. (2018) (both have mentioned and listed in the manuscript's reference) suggested the topographic beta spiral may provide a dynamic channel for subsurface Kuroshio cold water intrusion to the continental shelf. For detail, it may needs a numerical model to carry out. The purpose of this study is to present the valuable and precious data from the field measurements. The data presented in this study can provide for model calibration or verification to have better and correct understandings on the mechanisms.

We have revised the manuscript and marked the revisions by blue color.

---

## Author Comment (AC3) · 15 Jan 2019

The comment was uploaded in the form of a supplement:
https://www.earth-syst-sci-data-discuss.net/essd-2018-127/essd-2018-127-AC3-supplement.pdf

———————————————————————

---

## Author Response (AR2)

Comments to the Author:

Although the paper is not strongly supported by theories, the data are of a certain importance and the paper fulfil the requirements of ESSD. Its weakness is in the Introduction and an improvement is needed. These are some suggestions:

1) Page 2 – lines 24-25. The first sentence of the introduction is not nicely related to the next one. I understand the link, but this link is not really expressed. I suggest two possible solutions: the first is to delete the first sentence; the second is presented in bullet (3).
Reply: We accept the suggestion and have deleted the first sentence.

2) Page 3 – line 14: instead of 'during swimming, surfing, and snorkelling activities' I suggest the shorter 'leisure activities'.
Reply: We accept the suggestion and have revised the contents (marked by blue in the revised manuscript).

3) Page 5 – from line 4: The authors initially have introduced the effects of typhoons on temperature drops, and then present other possible mechanisms, including the intrusion of waters from Kuroshio. This kind of arrangement is creating some confusion and can explain the strong critics of referee 1. My suggestion is: at the beginning of the Introduction give a presentation of various mechanisms that can explain temperature drops (including the Kuroshio) and then start with the authors ideas on the effects of typhoons.
Reply:
We have added a paragraph to present the possible mechanisms that can explain temperature drops in the first section of Introduction, then described the importance of SST drop studies including its influences on marine ecosystem and human's activities in the ocean. After this first section, we present the previous studies on SST drop after typhoons (Section 2 of Introduction) including the significant cases in South China Sea (Section 3 of Introduction) and NE Taiwan waters (Section 4 of Introduction). In the last section of Introduction, we explain the motivation and objective of this study. I hope this organization helps the readers to understand the contents.

The remaining part of the paper can remain as it is.